# Supersonic propagation of lattice energy by phasons in fresnoite

M.E. Manley [1], P.J. Stonaha[1], D.L. Abernathy [2], S. Chi[2], R. Sahul[3], R.P. Hermann[1] & J.D. Budai [1]

Controlling the thermal energy of lattice vibrations separately from electrons is vital to many applications including electronic devices and thermoelectric energy conversion. To remove heat without shorting electrical connections, heat must be carried in the lattice of electrical insulators. Phonons are limited to the speed of sound, which, compared to the speed of electronic processes, puts a fundamental constraint on thermal management. Here we report a supersonic channel for the propagation of lattice energy in the technologically promising piezoelectric mineral fresnoite ($Ba_2TiSi_2O_8$) using neutron scattering. Lattice energy propagates 2.8−4.3 times the speed of sound in the form of phasons, which are caused by an incommensurate modulation in the flexible framework structure of fresnoite. The phasons enhance the thermal conductivity by 20% at room temperature and carry lattice-energy signals at speeds beyond the limits of phonons.

[1] Material Science and Technology Division, Oak Ridge National Lab, Oak Ridge, TN 37831, USA. [2] Neutron Scattering Division, Oak Ridge National Lab, Oak Ridge, TN 37831, USA. [3] Meggitt Sensing Systems, Irvine, CA 92606, USA. Correspondence and requests for materials should be addressed to M.E.M. (email: manleyme@ornl.gov)

A foremost scientific and technological challenge in energy sciences is to control the flow of heat carried by lattice vibrations[1]. The ultimate goal is to integrate thermal management with information processing, where heat in the form of lattice vibrations is not just directed away from hotspots as waste but is captured and converted into other forms for processing. To achieve this goal, it is crucial to understand the fundamental limits of lattice vibrations as a means to both capture and move heat. Spatial localization, or capture, of lattice vibrations is achievable by several known mechanisms, including nonlinearity-induced discrete breathers or intrinsic localized modes[2–5], disorder-induced Anderson localization[6–8], or confinement in nanostructures[9]. These mechanisms have all been explored recently as ways to impede lattice heat flow and enable efficient thermoelectric conversion[8–11]. At the other extreme, the upper limit for the speed of lattice vibrations is usually assumed to be the speed of sound. In theory, however, lattice vibrational energy can move much faster than sound—e.g. solitons in the one-dimensional Toda lattice always move faster than sound[12]— but there have been no observations of such supersonic purely lattice modes propagating in an insulating material in thermal equilibrium. The speed of sound is based on linear response theory, but solitons are nonlinear modes that exhibit locally augmented forces because of a compression of the lattice around the soliton. Solitons can warp the local environment in a way that enables them to travel faster than sound. While traveling solitons provide a possible mechanism for supersonic lattice energy transport, these tend to be limited to low-dimensional systems[13,14]. However, naturally occurring incommensurate structural modulations in crystals exhibit a related type of lattice excitation called a phason.

The existence of two or more mutually incompatible elements of translational symmetry necessitates the presence of continually accumulating phase shifts, which can be described in terms of a regular lattice of solitons (or walls)[15,16]. Using the classic analytical model of Frank and Van der Merwe[16], Fig. 1a–e illustrates how an incommensurate structure (Fig. 1a) develops from soliton-induced phase shifts (Fig. 1b) when formed in a regular soliton lattice (Fig. 1c). This modulation has the two types of dynamical modes illustrated in Fig. 1d. The amplitudon arises from excitations in the amplitude of the modulation (sharpness of solitons) and has optic-phonon-like dispersion (Fig. 1e). The phason is an excitation in the phase of the incommensurate modulation, and can be thought of as an acoustic phonon in the soliton lattice[16], where the soliton lattice takes the place of the crystal lattice. Phasons are gapless (Fig. 1e) because a uniform translation of the phase (soliton lattice) does not cost energy, just as acoustic phonons are gapless because a uniform translation of the crystal lattice does not cost energy. Notably, however, the response time of a soliton lattice can be faster than the linear response of the crystal lattice, hinting that phasons could be faster than acoustic phonons. However, the forces between the solitons (or walls) may also be weak, so it is not obvious a priori whether lattice phasons will be faster than acoustic phonons, and clear measurements of lattice phason propagation are needed. The incommensurate charge density wave in metallic blue bronze exhibits fast phasons[17], but these phasons carry charge and therefore do not provide a path to channel lattice heat separately from electrons.

The natural mineral fresnoite ($Ba_2TiSi_2O_8$), which has attractive properties for high-temperature piezoelectric sensor applications[18,19], exhibits a two-dimensional incommensurate structural modulation in the basal plane of its non-centrosymmetric tetragonal phase (space group $P4bm$)[20,21]. The fresnoite structure, illustrated in Fig. 1f, is unusual in that the titanium atoms are fivefold coordinated inside square pyramids[22].

The corner-linked titanium pentahedra and pyrosilicate groups form two-dimensional sheets perpendicular to the c-axis. The incommensurate modulation originates with rotations of the polyhedral within these sheets from one unit cell to the next, with an average periodicity that is incommensurate with the lattice[20,21].

Previous measurements of phasons in other systems have been challenging not just because of damping that broadens the mode[17,23–25], but also because the phasons appear in reciprocal space just below and often covered by the driving soft phonon mode that becomes the amplitudon[23].

In the following we show that the phasons in fresnoite are especially clear to observe because they do not sit directly below a soft phonon mode (amplitudon), as generally assumed[26,27], but rather appear at wavevectors directionally rotated away from the soft phonon modes. This allows for clear observations of the exposed phason dispersion and its temperature evolution, including the formation of highly supersonic phason dispersion cones. The phasons have group velocities ~2.8 times and ~4.3 times faster than longitudinal (LA) and transverse acoustic (TA) phonons, respectively. Our results demonstrate that the upper limit for the propagation of pure lattice energy is well above the velocity of sound.

## Results

**Time-of-flight inelastic neutron scattering**. The neutron scattering measurements in Fig. 2 were obtained using the ARCS time-of-flight spectrometer at the Spallation Neutron Source (see Methods), and provide a broad overview of changes in both the static and dynamical structure of fresnoite upon crossing the incommensurate modulation transition temperature (~433 K). Figure 2a shows the elastic scattering in the (HK0) basal plane above the transition at 485 K. At this temperature there are no incommensurate superstructure reflections but rather a zigzag pattern of weak thermal diffuse scattering around the (H, −7, 0) reflections. Since these cuts near zero energy transfer are meant to remove the phonons, thermal diffuse scattering is not expected unless there are very low-energy modes dipping into the energy-resolution-limited elastic scattering. An energy-momentum slice through the thermal diffuse pattern along $\mathbf{Q} = [H, -6.65, 0]$ in Fig. 2b reveals soft optic phonon modes at these locations. The soft phonon modes in the middle of Fig. 2b near $\mathbf{Q} \approx (\pm 0.4, -6.65, 0)$ and those near $\mathbf{Q} \approx (\pm 1.4, -6.65, 0)$ are all symmetry equivalent. The $E = 2.5 \pm 0.5$ meV constant-energy slice in the (HK0) plane in Fig. 2c reveals the positions of the soft phonon modes and the acoustic phonon dispersion cones. The soft modes in Fig. 2c correlate well with the thermal diffuse scattering in the elastic scattering in Fig. 2a. The elastic scattering in the (HK0) plane below the transition at 381 K is shown in Fig. 2d. Incommensurate superstructure reflections are clearly visible at this temperature. Contrary to what is expected from soft mode theory[27], the positions of the incommensurate superstructure reflections do not correlate with the soft phonon mode positions. Figure 2e shows the same energy-momentum slice as Fig. 2b below the transition. While soft phonon columns of intensity at $\mathbf{Q} \approx (\pm 0.4, -6.65, 0)$ near 2 meV diminish, they are replaced by steeply dispersing features that extend up from the incommensurate superstructure reflection positions. These features are the phasons expected with the incommensurate modulation, as indicated in Fig. 2e. Figure 2f shows the same constant-energy slice at $E = 2.5 \pm 0.5$ meV in the (HK0) plane as Fig. 2c below the transition at 381 K. In addition to the slice through the acoustic phonon cones above the primary reflections, much smaller intensity circles can be seen from cutting through the phason dispersion cones. The much smaller diameter phason circles

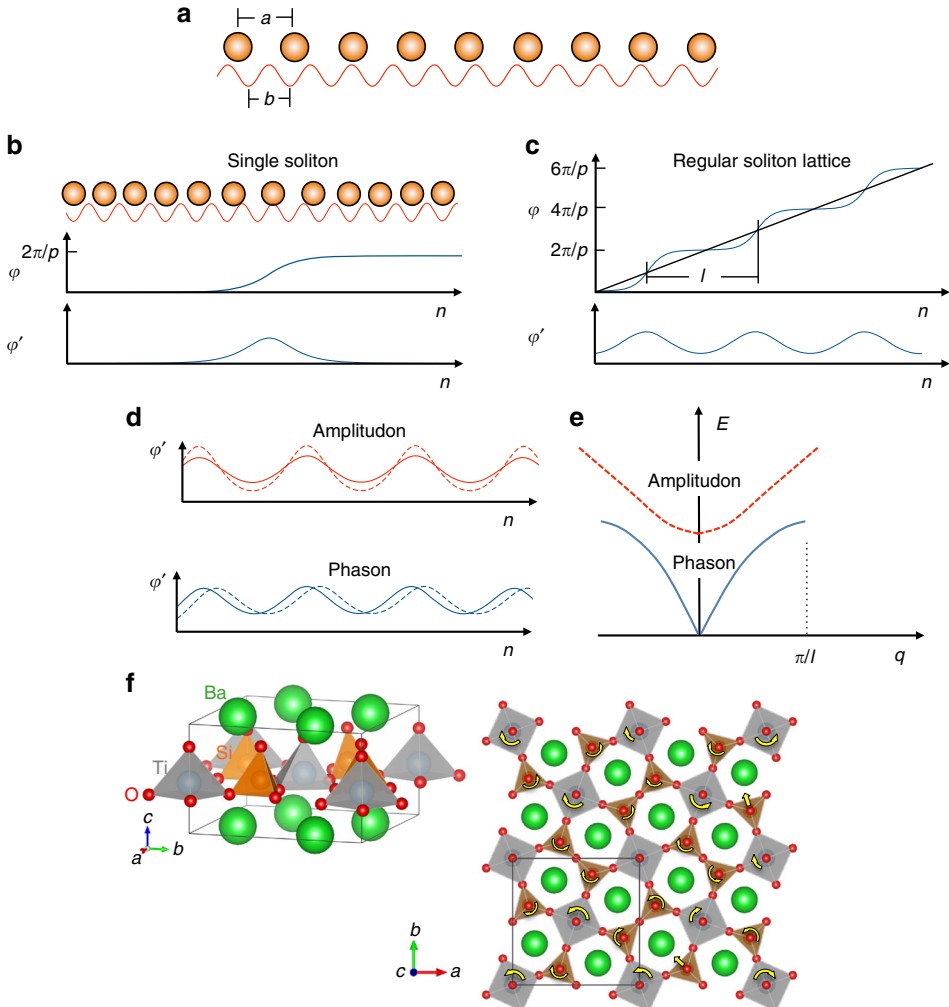

**Fig. 1** Phasons, amplitudons, and the structure of fresnoite. **a** An incommensurate structure in one dimension, where the two lattice parameters, $a$ and $b$, have a non-integer ratio. **b** A single soliton (or wall) separating two commensurate structures. The single soliton solution after ref. [16] stretches the lattice and introduces the phase shift, $\varphi = 2\pi/p$, which separates the two commensurate structures ($\varphi'$ is the derivative of the phase with respect to the site coordinate, $n$, which is taken to be continuous in this model[16]). **c** A regular soliton lattice of spacing $l$ introduces the continuous phase shifts of the incommensurate structure in **a**, where the straight line represents the unperturbed incommensurate structure[16]. **d** Illustration of the amplitudon and phason excitations in the soliton lattice. **e** Dispersion curves of amplitudon and phason excitations. **f** Tetragonal crystal structure of fresnoite. Below ~433 K a modulation develops primarily with rotations of the tethered polyhedral in the basal plane, which occur from one unit cell to the next with a periodicity that is incommensurate with the lattice[20, 21]. The yellow arrows indicate a subset of rotations and translations in the modulation. Note how propagating a rotation around a ring of five polyhedral forces the translation of one of the Si pyramids near the bottom center. Such geometric frustration leads to complex structure

indicate that the phasons emerge from the incommensurate-superstructure reflections with much steeper dispersion. This shows that the phason velocities are highly supersonic. The circular shape also reveals that the phasons are isotropic in the ($HK0$) basal plane. Furthermore, the positions of the soft phonons and the phasons themselves form a larger half circle in the zone (Fig. 2f), indicating that they are related by a wavevector rotation (Fig. 2g).

**Triple-axis inelastic neutron scattering**. To obtain a more detailed look at the phason dispersion and temperature dependence we use triple-axis neutron scattering. The data shown in Fig. 3 were obtained using the HB3 triple-axis spectrometer at the High Flux Isotope Reactor (see Methods), and provide detailed information on phason formation and dispersion near the incommensurate superstructure reflection at $\mathbf{Q} \approx (0, -5.4, 0)$. At

high temperature, Fig. 3a, there is no incommensurate superstructure reflection, but there is inelastic intensity extending between the point where the reflection eventually forms and the TA phonon near 7 meV. With decreasing temperature (Fig. 3b–f) the inelastic intensity gradually fills in this region to form a phason branch. We label this branch the T-phason (Fig. 3f) since this measurement is in transverse geometry. The instrument resolution ellipsoid, shown as an inset in Fig. 3e, is aligned with the T-phason propagating in the positive $H$ direction, which emphasizes the dispersion for positive values of $H$. The width of the T-phason dispersion curve is near the resolution limit at 387 K. The shape of the dispersion curve does not change significantly with temperature. Figure 3g, h compare the temperature dependence of the integrated intensities of the incommensurate reflection at $E = 0$ and the phason inelastic scattering at $E = 2.5$ meV. At 469 K the T-phason has 25% of its intensity at 387 K, while the incommensurate-superstructure reflection has only

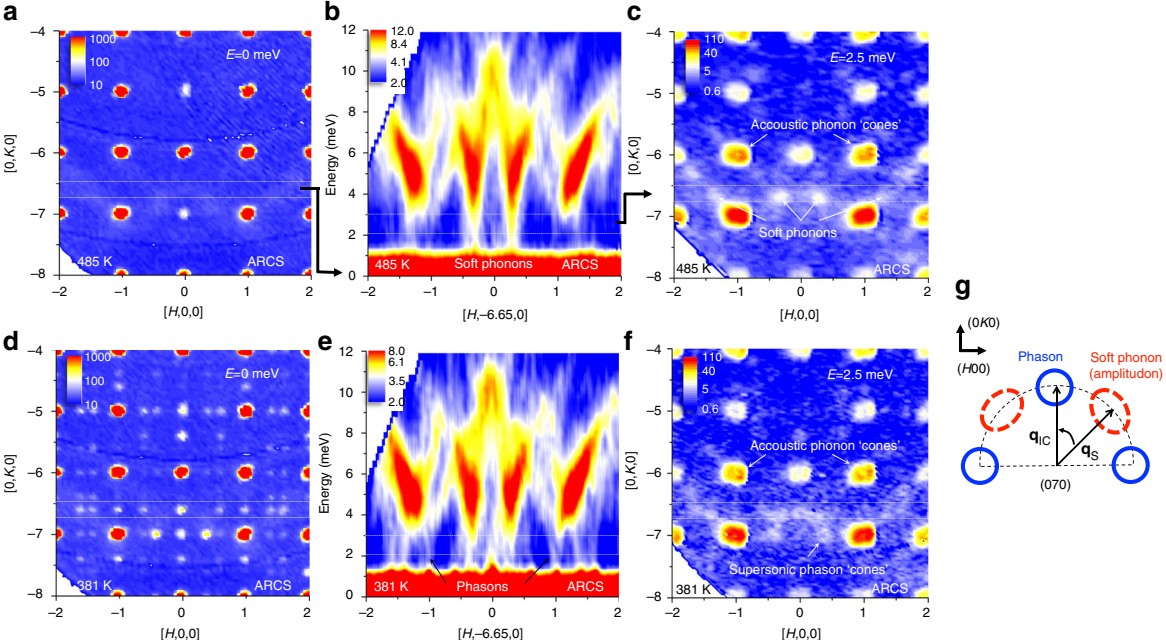

**Fig. 2** Dynamic structure measured on a time-of-flight spectrometer. **a** Elastic slice ($E = 0 \pm 0.5$ meV) measured above the incommensurate transition temperature ($T_C \sim 433$ K) transition at 488 K in the (HK0) plane. **b** Inelastic scattering along $\mathbf{Q} = [H, -6.65, 0]$ showing soft optic modes at 488 K. **c** Constant energy slice ($E = 2.5 \pm 0.5$ meV) in the (HK0) plane at 488 K. **d–f** Show the same slices as **a–c** respectively, but measured below the transition at 381 K. **g** Illustration of the relationship between the phason wave vector, $\mathbf{q}_{IC}$, and the soft phonon (amplitudon) wave vector, $\mathbf{q}_S$

about 0.5% of its intensity at 387 K. Furthermore, the growth of the T-phason intensity slows for temperatures below about 400 K (Fig. 3h), while the incommensurate-superstructure reflection continues to grow rapidly down to 250 K (Fig. 3g). This indicates that the phasons are gapped ahead of the static incommensurate modulation. While ideally phasons are expected to be gapless since the phase energy is translational invariant for true incommensurate modulations, small gaps are possible if the phase is pinned by crystal defects[16,23] or by any mechanism that breaks translational invariance, including a high-order commensurate modulation[28]. For example, if the soliton thickness is similar to the lattice spacing it may tend to lock-in to a site in the unit cell, resulting in the phase not being strictly incommensurate and the phasons being pinned[28]. The closing of the gap (growth of incommensurate elastic peak) indicates a more complete incommensurability develops at lower temperatures.

**Analysis of the lattice phason properties**. Qualitatively distinguishable features of the dispersion, thermodynamic, and anisotropic thermal transport properties shown in Fig. 4 support the identification of these features as phasons rather than acoustic phonons. Figure 4a compares the phason dispersion with that of the TA and LA phonons in the (HK0) basal plane. The phason was measured in both transverse (T-phason) and longitudinal (L-phason) geometries in the basal plane and from several incommensurate superstructure reflections. In contrast to the acoustic phonons, there is no significant difference in the dispersion of the T-phason and L-phason (the phasons are shifted to the zone center for comparison to the acoustic phonons). The shapes of the phason dispersion curves are similar to the acoustic modes, as expected[16,23], but the phasons are much steeper at low energies. At 2 meV the T-phason is ~4.3 times steeper than the TA phonon and the L-phason is ~2.8 times steeper than the LA phonon, confirming that at low energies the phasons propagate at highly supersonic velocities. The fresnoite heat capacity[29] divided by temperature cubed ($C_P/T^3$) shown in Fig. 4b exhibits a clear deviation from the Debye $T^3$ law. While at high temperatures the

heat capacity is well described by phonon theory, below 3 K there is a clear upturn in $C_P/T^3$ rather than the constant expected from the Debye $T^3$ law. This is a known feature of the heat capacity of incommensurate structures, and has been explained in terms of phason damping or possibly a small gap from phason pinning at long wavelengths ($q = 0$)[30]. Unlike acoustic phonons, phasons are always overdamped at long wavelengths because a uniform sliding of the modulation phase has viscosity[30]. Accounting for this $q = 0$ phason damping or pinning in the heat capacity calculation produces upturns in $C_P/T^3$ similar to that observed in Fig. 4b[30]. Hence, the thermodynamic properties are consistent with the presence of phasons in fresnoite.

The anisotropic thermal diffusivity of fresnoite[19], shown in Fig. 4c, reveals a difference between the phason propagation direction (a-axis) and the direction perpendicular to the phason propagation direction (c-axis). Along the c-axis the thermal diffusivity decreases gradually with temperature, except for a small dip associated with the incommensurate transition temperature ($T_C \sim 433$ K). Along the a-axis, by contrast, the thermal diffusivity increases markedly below the temperature at which phasons form. This is similar to what is described in incommensurate charge density wave systems at low temperatures and attributed to charge-density-wave phasons of rather large velocity[31]. Attributing the rise in fresnoite to lattice phasons (blue area in Fig. 4c), we estimate that phasons increase the in-plane thermal diffusivity by about 20% at room temperature, and this contribution will likely continue to grow with further cooling. There is also an associated increase in the anisotropy (a-axis/c-axis) from 1.8 at 870 K to 2.7 at 294 K, and this is also expected to increase with further cooling.

The relative magnitude of the phason contribution to thermal transport can also be estimated by analyzing the components of the Boltzmann transport equation. The diagonal components of the thermal conductivity tensor are[32]

$$\kappa_{\alpha\alpha} = \frac{1}{V} \sum_{q_s} C_{q_s} v_{\alpha q_s}^2 \tau_{\alpha q_s}, \tag{1}$$

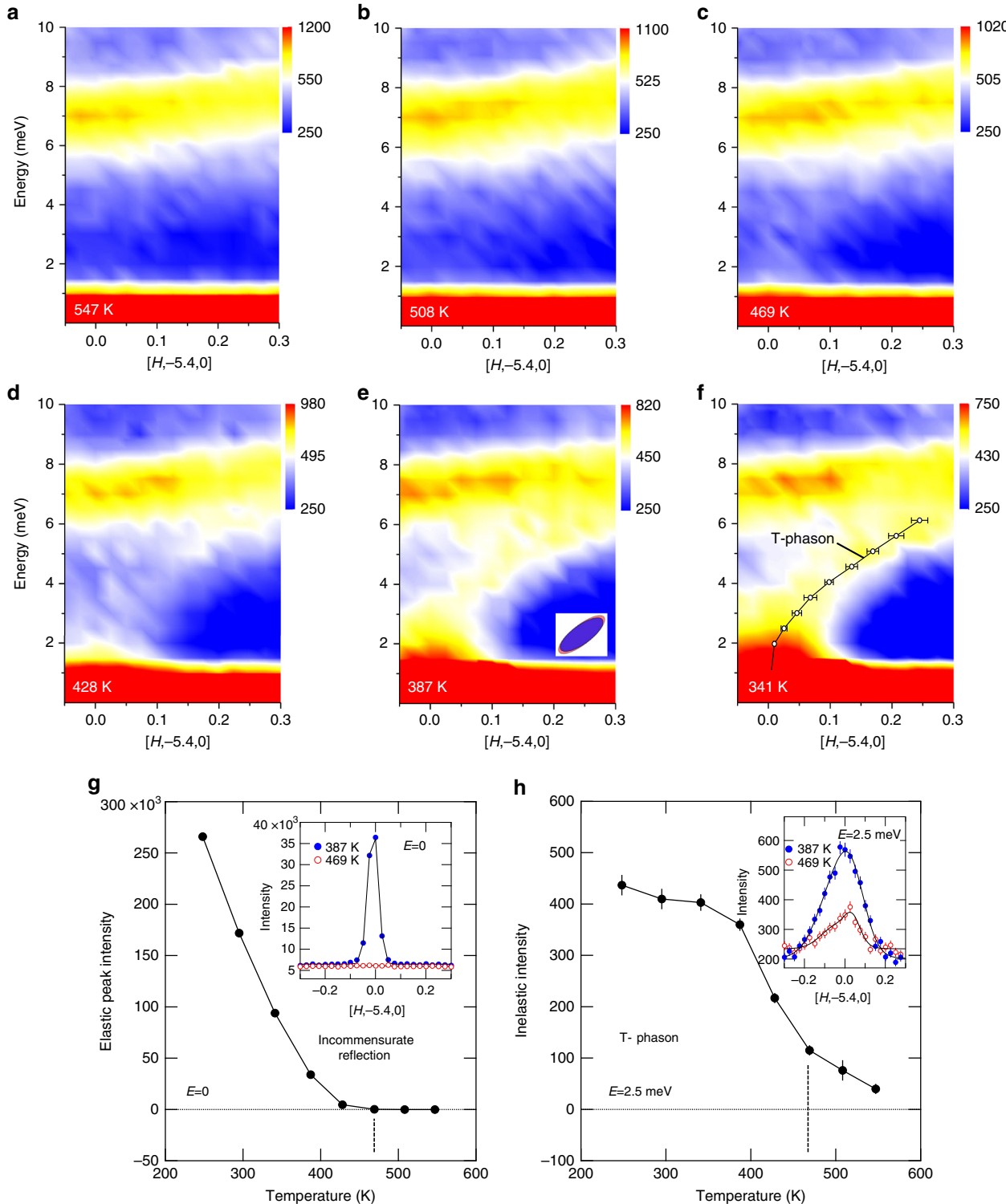

**Fig. 3** Phason formation measured on a triple-axis spectrometer. These measurements provide a detailed assessment of the temperature dependence of the phason and incommensurate reflection in a focused region of **Q**-$E$ space along **Q** = [$H$, −5.4, 0]. **a–f** Formation of the (0, −5.4, 0) phason on cooling below the incommensurate phase transition. The tilted ellipsoid inset in **e** indicates the instrument resolution function. **g** Shows the temperature dependence of the incommensurate reflection intensity measured at the elastic line ($E = 0$), and **h** shows the temperature dependence of the phason intensity measured at $E = 2.5$ meV. Uncertainties are statistical and represent one s.d.

where $v_{\alpha q_s}$ is the group velocity of the mode $q_s$, $\tau_{\alpha q_s}$ is the mode lifetime for transport in direction $\alpha$, $C_{q_s}$ is the mode heat capacity, and $V$ is volume. At high temperatures ($k_B T \gg \hbar\omega$) the heat capacity of each phonon branch contributes $1k_B$ per primitive cell volume. The acoustic phonons normally dominate the thermal

conductivity and there are 2 transverse and 1 longitudinal modes, making for a total of $3k_B$ per primitive cell. The phason heat capacity is smaller because it comes from a degree of freedom that is shared with a larger superstructure. In the basal plane the superstructure is incommensurate with reflections near $h \sim 0.4$,

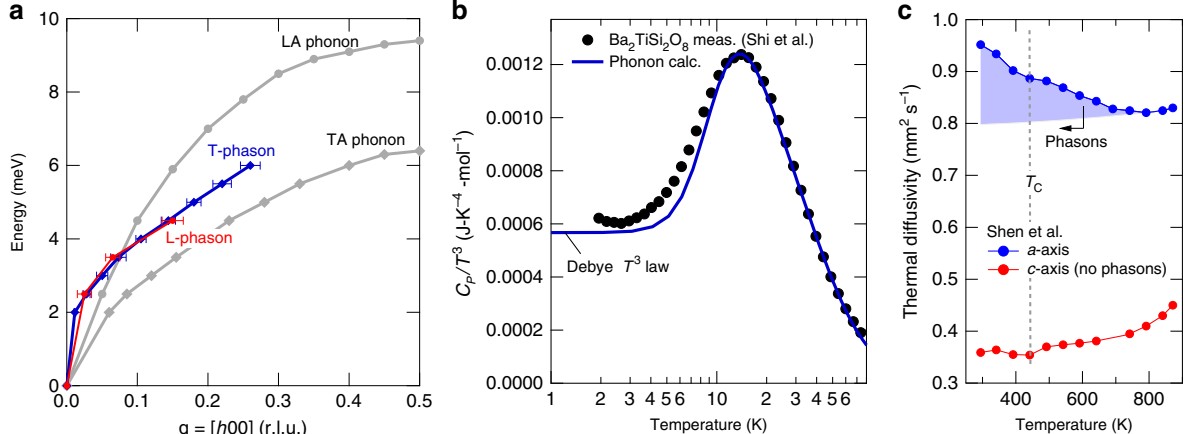

**Fig. 4** Phason dispersion, heat capacity, and thermal diffusivity. **a** A comparison of the phason dispersion with that of the transverse acoustic (TA) and longitudinal acoustic (LA) phonons. The T-phason was measured in transverse geometry and the L-phason was measured in longitudinal geometry. In contrast with acoustic phonons, the phasons show no significant difference when measured in transverse or longitudinal geometry. **b** Heat capacity divided by $T^3$ as a function of temperature for fresonoite. The measured black data points are from Shi et al.[29]. The blue curve is calculated from a phonon model that includes a Debye term, with a Debye temperature of $\theta_D = 354$ K, plus Einstein modes positioned at the Van Hove singularities determined from the measured acoustic phonons (5.5 and 6.2 meV for TA modes, and 9.5 meV for LA modes). The peak around 14 K comes primarily from the Einstein modes at 5.5 and 6 meV representing the flat portions of the TA modes near the zone boundaries (see **a** for the in-plane TA and LA modes). The flat region at lowest temperatures comes from the Debye term. **c** Anisotropic thermal diffusivity as a function of temperature for fresnoite after Shen et al.[19]. Phasons propagate in the basal plane along the a-axis (blue symbols) but not out of the plane along the c-axis (red symbols). The incommensurate modulation transition at $T_C \sim 433$ K is indicated by the gray dashed line. The shaded blue region is an estimate of the phason contribution. Uncertainties are statistical and represent one s.d.

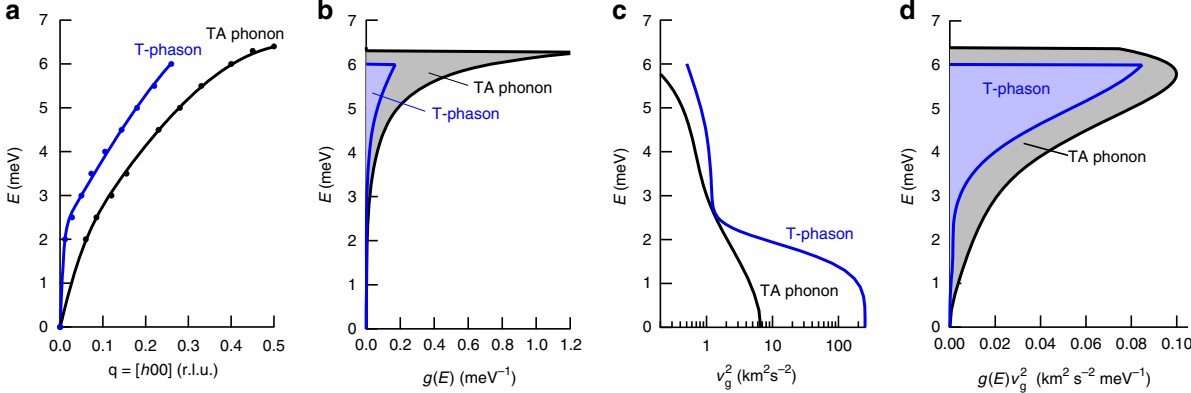

**Fig. 5** Phason and phonon thermal conductivity from dispersion. **a** Analytic curves fit to the dispersion data. Both data sets were fit using the function $E(q) = A\sin\alpha q + B\tanh\beta q$, where the fit constants are $A = 5$ meV (5 meV), $\alpha = 2.8$ r.l.u.$^{-1}$ (3.38 r.l.u.$^{-1}$), $B = 1.447$ meV (2.166 meV), and $\beta = 17.27$ r.l.u.$^{-1}$ (103.9 r.l.u.$^{-1}$) for the TA phonon (T-phason). **b** Mode densities of states, $g(E)$, for the TA phonon normalized to 1 and the T-phason normalized to 0.16 (estimated from incommensurate superstructure, see text). The mode partial densities of states scale as $g(E) \propto q^2(E)(dq/dE)$. **c** Square of the mode group velocities $v_g^2 = (dE/dq)^2$. **d** Density of states weighted squared group velocities. Assuming similar mode lifetimes the blue area relative to the gray area in **d** estimates a T-phason contribution that is about 50% of the TA phonon contribution

which corresponds to a cell size increase of ~2.5. Therefore, the cell volume increases by $\sim 1 \times 2.5 \times 2.5 = 6.25$, and the phason contribution to the density of states is about 1/6.25 (16%) of the contribution from acoustic phonons. However, the higher phason group velocities, which are squared in the expression for thermal conductivity (Eq. (1)), increase the phason contribution to the thermal conductivity. Figure 5 shows, based on the fits to the dispersion curves (Fig. 5a), that the smaller calculated T-phason density of states (Fig. 5b) when multiplied by the calculated higher squared phason group velocities (Fig. 5c) results in a contribution that is similar to the TA phonon (Fig. 5d). The density-of-states weighted square velocity integrates to an area (blue region in Fig. 5d) that is 50% of that of the TA phonon

(gray region in 5d). This means that for similar excitation lifetimes and fully occupied states the T-phason contribution to thermal conductivity is expected to be 50% that of the TA phonon. It is therefore not surprising that the still growing phasons already contribute 20% of the total in-plane thermal conductivity at room temperature (Fig. 4c). This analysis neglects unknown differences in the mode life times. The acoustic phonons and phasons have measured line widths that are instrument resolution limited, so we cannot resolve differences in lifetime broadening directly. The known damping of long wavelength phasons[30] should not change this estimate of the thermal conductivity, however, since the density of states at long wavelengths is very small (Fig. 5). While the phason contribution

to bulk transport is important, a more direct application would be to use fast phasons as a signal in a thermal logic device[1], although it remains to be seen whether phason detection can be made practical.

## Discussion

Apart from supersonic propagation velocity, the most interesting characteristic of the phasons in fresnoite is their separation from the soft phonon mode. Normally with such transitions, a soft phonon mode first drops towards zero energy as the transition temperature is approached, signaling a structural instability at the soft mode wave vector, $\mathbf{q}_s$. Below the transition, the soft phonon splits into two modes at this same wave vector, the amplitudon and the phason[28]. Fresnoite does exhibit a soft phonon mode that drops towards zero energy, signaling a structural instability at the soft mode wave vector, $\mathbf{q}_s$. However, the phasons and incommensurate superstructure reflections occur at different wave vectors, $\mathbf{q}_{IC}$. The relationship between the soft mode $\mathbf{q}_s$ and the phason $\mathbf{q}_{IC}$ is a rotation of the wave vectors with the same magnitude from [110] directions to [100] directions, as illustrated in Fig. 2g. This can also been seen in the half-circle arrangement of the phason and soft phonon (amplitudon) intensity around the $\mathbf{Q} = [0, -7, 0]$ point in Fig. 2f. This result is surprising since intuitively a soft phonon mode, which indicates instability to modulate at $\mathbf{q}_s$, would predict an incommensurate superstructure modulation at the same wave vector. Rather, here the soft mode correctly predicts the wavelength of the modulation but not the direction. This wave vector rotation is reminiscent of the so-called $\mathbf{Q}$-rotation effect that occurs with the helical magnetic spin-density wave in MnSi[33–35]. In this case, the wave vectors of the spin-density wave rotate towards the direction of an applied magnetic field, but the magnitude of the wave vectors do not change. This rotation has been explained in terms of wave-vector reorientation phase transitions arising from instability in the phason strain field[36]. Phason strain instability may also drive the phason rotation in fresnoite. In light of fresnoite's piezoelectricity[18,19], it would be interesting to explore whether electric-field-induced strains produce a $\mathbf{Q}$-rotation effect or in some way modify the supersonic phasons. Melilite, a related piezoelectric mineral with a similar incommensurate modulation[37], has modulation wave vectors (phasons) that vary linearly with temperature[38]. At the other extreme, it would also be interesting to explore phasons in mullite, which has a very stable (rigid) incommensurate structure that persists to higher temperatures, all the way until melting[39–41].

Summarizing, our results reveal highly supersonic propagation of pure lattice energy in the naturally occurring mineral fresnoite in thermal equilibrium, breaking the conventional limit set by the speed of sound by factors ranging from 2.8 to 4.3. The supersonic phasons carrying this thermal energy are exposed in our neutron scattering measurements by a wave vector rotation that moves the phason dispersion cones away from interference from the soft phonon mode. This rotation challenges established ideas on how incommensurate structural modulations develop from soft phonon modes and suggests unexpected phason reorientation instability. Taken together these remarkable findings open a new venue for understanding and controlling the transport of lattice energy beyond the limits of phonons.

## Methods

**Time-of-flight inelastic neutron scattering**. Time-of-flight neutron scattering measurements were performed on a ~50 cm³ fresnoite crystal ($Ba_2TiSi_2O_8$) using the ARCS at the Spallation Neutron Source, Oak Ridge National Laboratory[42]. The crystal was mounted in a furnace with the $(HK0)$ reflections in the scattering plane using a vanadium holder. Measurements were performed at 485 K (above the 433 K incommensurate transition temperature) and at 381 K (below incommensurate transition temperature) with the crystal oriented in the $(HK0)$ plane and with

incident neutron energies of 25 meV. Four-dimensional $\mathbf{Q}$-$E$ volumes of data were obtained by rotating the angle between the [100] axis and the incident beam in 0.5° steps and combining the data using the HORACE software package[43]. Data were collected at each angle from −20° to 70° to obtain a complete data set in the $(HK0)$ plane.

**Triple-axis inelastic neutron scattering**. The same fresnoite crystal ($Ba_2TiSi_2O_8$) was also measured on the HB3 spectrometer at the High Flux Isotope Reactor at Oak Ridge National Laboratory. The HB3 spectrometer was operated with a filtered fixed final neutron energy of 14.7 meV with horizontal collimation of 48':40':60':120'. Pyrolytic graphite PG(002) was used for both the monochromator and the analyzer. The crystals were mounted in a furnace with the $(HK0)$ reflections in the scattering plane using the same vanadium holder. Measurements were performed in constant energy mode along $\mathbf{Q} = [H, -5.4, 0]$ at energies from 0 to 10 meV in 0.5 meV steps. The scans were repeated at different temperatures including 248, 295, 341, 387, 428, 469, 508, and 547 K (the incommensurate transition for $Ba_2TiSi_2O_8$ is ~433 K).

**Crystal growth and characterization**. Oxides of Ti and Si and $BaCO_3$ were mixed together in stoichiometric ratio and sintered using solid-state reaction to form $Ba_2TiSi_2O_8$ precursor material. A fresnoite single crystal was grown using conventional Czochralski crystal growth technique. The precursor material is placed in a crucible and melted above its melting point (~1450 °C). The <001> seed crystal is mounted on a pull rod and lowered to surface contact with the melt layer to initialize seeding. As the pull rod is slowly raised away from the melt, the crystallization occurs continuously as the surface layers cools below the melting point. The linear displacement of the pull rod and the temperature of the crucible are controlled very finely to reduce thermal cracks during the crystallization. The crystal was successfully grown in the $[001]_c$ direction. The as-grown crystal was oriented using x-ray diffraction. The tetragonal Fresnoite crystal with 4 mm symmetry has the following piezoelectric coefficients:

$$\begin{pmatrix} 0 & 0 & 0 & 0 & d_{15} & 0 \\ 0 & 0 & 0 & d_{15} & 0 & 0 \\ d_{31} & d_{31} & d_{33} & 0 & 0 & 0 \end{pmatrix}. \tag{2}$$

The piezoelectric properties were measured using the IEEE standard method for measuring full property data of piezoelectrics. The longitudinal coefficient ($d_{33}$) was measured to be ~4 pC/N, as expected[19].

**Data availability**. The data that support the findings of this study are available from the corresponding author on request.

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

## Acknowledgements

Research sponsored by the U.S. Department of Energy, Office of Basic Energy Sciences, Materials Sciences and Engineering Division. This research used resources at the High Flux Isotope Reactor and Spallation Neutron Source, a DOE Office of Science User Facility operated by the Oak Ridge National Laboratory.

## Author contributions

M.E.M. conceived of the experiments. M.E.M., P.J.S., D.L.A., and J.D.B. performed the time-of-flight neutron scattering measurements. M.E.M., S.C., P.J.S., and R.P.H. performed the triple-axis neutron scattering measurements. M.E.M. analyzed the data. R.S. grew and oriented the single crystal, and characterized its piezoelectric properties. M.E. M. wrote the manuscript with input from R.P.H., D.L.A., J.D.B., and S.C.

## Additional information

**Competing interests:** The authors declare no competing interests.

