## [Peer Review File · Nature Communications]

Reviewers' comments:

Reviewer #1 (Remarks to the Author):

This paper is a very valuable contribution to the understanding of incommensurate materials.

I do have some concerns regarding the validation of homogeneity of the 'crystal' from which phason measurements were taken. This is because fresnoite crystals can prove more complex at nanoscale than X-ray diffraction measurements indicate. My key queries are:

1. Is there an independent verification of the stoichiometry? In particular, have chemical analyses been undertaken to ensure the Ba:Ti:Si = 2:1:2 stoichiometry is correct without Ba vacancies?
2. Has TEM, high resolution backscattered SEM, or EBSD channeling of [001] sections been used to confirm the absence of micron sized domains.

It would have been useful to see if the supersonic performance of the crystal was maintained after temperature cycling through the incommensurate transition temperature. Two cycles would be sufficient.

In general, mellites are more homogeneous than fresnoites and perhaps this could be commented upon.

Reviewer #2 (Remarks to the Author):

The manuscript by Manley et al. reports neutron scattering measurements of a fresnoite crystal, where branches of phasons were discovered as a result of the development of incommensurate superstructures below a transition temperature of 433K. By comparing the neutron scattering intensity maps above and below the transition temperature, the development of the incommensurate superstructure was clearly shown as emerging Bragg reflection points, around which dispersion cones of phasons were observed. Two interesting features were noted and discussed: the supersonic group velocity of phasons and the mismatch of the wavevectors between the phasons and the soft phonon mode. The former is of high interest to thermal energy transport, and the latter is explained by phason strain instability possibly induced by the piezoelectricity. The experimental observation of the phason dispersion is remarkable, and as far as this reviewer is concerned, the experimental evidence is solid and convincing. The finding of supersonic phasons is of interest to thermal transport, and is expected to stimulate follow-up investigations, for example how electric field can affect the phason properties. In this light I can recommend publication of this manuscript in Nature Communications, provided the following minor comments are addressed:

1. It would greatly benefit a general audience if more background information could be given in the introduction part, in addition to citing the results of Ref. 16, given that the concept of phason is unfamiliar to the thermal transport community. For example, what is the physical picture (lattice vibration modes) of a propagating phason? what determines the velocity of a phason, and why its velocity can exceed the sound velocity? How much energy can the phasons carry, in comparison to phonons?
2. In addition to discussing the heat capacity in Fig. 4b, it'd be helpful to also look at the thermal conductivity data, if it is available, to see whether there are signs of contribution from the phasons.
3. In Fig. 1c, the legend "regular soliton lattice" is not correctly displayed. It would be helpful to provide color bars for Figure 2 and Figure 3. In Figure 1f, the incommensurate modulation is not clearly seen; it would be educational to clearly mark the incommensurate rotations of polyhedral.
4. Line 142-145, it is unclear why the slowdown of the growth of the t-phason intensity with decreasing temperature below 400K indicates the t-phason is partly gapped. More explanation and background information would be helpful.

5. It is noted in the introduction that previous measurements of phasons suffered from damping that broadened the mode. It would be helpful to also discuss the role of phason damping in this work.

Reviewer #3 (Remarks to the Author):

This is an interesting paper that reports on the observation of a phason mode in the piezoelectric mineral fresnoite by inelastic neutron scattering (INS) experiments. The standard theory of displacive incommensurate (IC) phase transitions as it has been formulated nearly 40 years ago, goes as follows: at some point a phonon mode becomes critical and softens at an IC reciprocal space position to finally condensing at a given temperature. Below the critical temperature a new periodicity appears and the soft mode is further decomposed into modes that account for variations of the phase and of the amplitude of the modulation. Nevertheless the occurrence of this decomposition soft phonon \rightarrow phason + amplitudon (as nicely shown in the cartoon of Figs 1d and 1e) is not often observed in INS experiments for reasons that are still under debate. It is always nice to have one more compound where these ideas can be further tested, and I congratulate the authors for this new study.

This compound offers at least two interesting features readily highlighted by the authors : (1) the transition temperature of this IC modulation is rather high, 433K, and (2) the group velocity of the phason mode is larger than the normal longitudinal speed of sound by a factor of 3. Implicit in this statement is the fact that because of the presence of this IC structure, and thus of phasons with "supersonic" velocities, in fresnoite at high temperature this property can be used to dissipate thermal energy more efficiently and overcome the limits of normal acoustic phonons. It is here where this reviewer thinks that the authors have made some conceptual mistakes and where the experimental support of their hypothesis is terribly lacking.

My mayor concerns are :

1.- The manuscript reports inelastic neutron scattering and not thermal conductivity experiments. Without these latter experiments the conclusions announced from the title of this manuscript can not be supported and remains very speculative. In addition, as I detail below, there are good reasons to believe that phasons do not serve for lattice energy transport purposes.

2.- The occurrence of a phason mode in this compound is very likely beyond dispute, at least this reviewer does not dispute it. However it has not been experimentally shown whether the phason mode is gapped or gapless. In page 5 the authors claim it is gapped (I do not understand the term "partly gapped") and a few lines below they hold that the gap closes at lower temperatures (!). No experimental evidence is provided for any of these assertions. The authors know that if they want to properly assess the phason gap condition and thus help disentangle this problem INS experiments with cold neutrons, that warrant a better energy resolution, are mandatory. Experiments in different IC compounds have shown that the phason mode always (?) has a gap or at least it is overdamped (like in K_2SeO_4). I believe the authors are acquainted with the book by Blinc and Leyanyuk "IC phases in dielectrics" where they can found an early account of the phason gap features in several compounds. As the authors have stated the C_p/T^3 plot shows an upturn at low temperatures, although there is no clear signature of a phason gap, in much the same way as the results displayed in Fig. 4. This upturn is very likely produced by the damping of the phason gap. A recent account of this kind analysis in IC $ThBr_4$ can be found in Remenyi et al., PRL 114, 195502 (2015), following the ideas in ref. 30. The presence of a phason gap and damping hinders the thermal conductivity. In view of this and of the presence of an upturn of the CP/T^3 attributed to phason damping, this reviewer does not understand why the authors still believe the title of this manuscript !

3.- In order for the phason to be relevant for lattice transport properties an order of magnitude of

its contribution to the density of states should be provided. The crucial point is to understand that the phason and amplitudon modes just appear in the close neighbourhood of the IC position, and that the volume thus encapsulated is relatively small compared to the volume of the Brillouin zone. Put in a different way, if there are $3N$ vibration modes, N the number of atoms in the unit cell, each mode contributes as 1. In the case of phasons this number should be much smaller, in many cases its contribution falls down to 0.01-0.1. You can check this issue in, for instance, the paper by Boriack & Overhauser, PRB18, 6454 (1978). This argument on the smallness of the volume of the phason excitation stands irrespectively of whether the phason is gapped or gapless.

3a.- If the phason is gapped one should expect a deviation of the C_p/T^3 curve, hardly visible because of the smallness of its contribution. There has been some heated discussions in relation to the contribution of the phason to the low temperature properties of the heat capacity. I wonder if the authors are acquainted with this. Figure 4 shows specific heat data with a phonon calculation unable to fit the data below 10 K. May be this a gapped phason contribution ?

3b.- If the phason is gapless one should expect that its contribution to C_p goes to slightly decrease (!) the Debye temperature from regular acoustic phonons. And again this correction should be very very small. Again it should be interesting to put some numbers on the renormalisation of the Debye temperature by gapless phasons.

4.- Most of the heat transport is conducted by transverse acoustic phonons at all temperatures. The reason for that is that the heat transport is proportional to the phonon density of states (PDOS) and the transverse acoustic phonons have a larger PDOS than the longitudinal ones. Any "new" mode with larger group velocity should renormalise the PDOS, and thus the Debye temperature, to lower values. I am afraid that this has not been checked either.

5.- Thermal conductivity (TC) is rather difficult to calculate from first principles. On general grounds the TC evolution as a function of temperature goes as follows. At low temperatures, when the boundary scattering mechanism dominates, the conductivity follows the expected increase as T^3 . As the temperature increases, mass-defect scattering becomes important and reduces the T^3 dependence. Three-phonon scattering sets in at finite temperatures, becoming dominant at high temperatures. As a result, the conductivity reaches a maximum (the position of the maximum goes from 20K in Si to 100K in C-diamond) before starting to decrease. There are four criteria for choosing high-thermal-conductivity materials: (i) low atomic mass, (ii) strong interatomic bonding, (iii) simple crystal structure, and (iv) low anharmonicity. Conditions (i) and (ii) help increase the Debye temperature, condition (iii) means a low number of atoms per unit cell, resulting in fewer optical branches and hence fewer anharmonic interactions, and condition (iv) means reduction in the strength of anharmonic interactions. This compound may satisfy the first, hardly the second criteria but not at all the third and fourth.

Summarising, the manuscript contains important information on IC dynamics, in particular the wavevector rotation away from the soft mode position, that is worth publishing. However this reviewer can not recommend this manuscript for publication in Nature Communications. Indeed, each one of these above mentioned 5 points invalidates the authors view on the possibility of supersonic transport of the thermal energy of lattice vibrations by phasons, which is the message that would warrant publication in this journal.

Response to referees

NCOMMS-17-23667A-Z: "Supersonic transport of lattice energy by phasons in fresnoite"

Reviewer #1

Comment #1 of Reviewer #1: *This paper is a very valuable contribution to the understanding of incommensurate materials.*

Reply: We thank the referee for this positive assessment.

Comment #2 of Reviewer #1: *I do have some concerns regarding the validation of homogeneity of the 'crystal' from which phason measurements were taken. This is because fresnoite crystals can prove more complex at nanoscale than X-ray diffraction measurements indicate.*

Reply: The time-of-flight neutron scattering measurements done on ARCS are sensitive to both nanoscale inhomogeneity and local atomic displacements. The energy resolution allows us to separate out the lattice vibrations (inelastic thermal diffuse scattering) from the elastic diffuse scattering of static inhomogeneity, which is a distinct advantage over x-rays. We have experience observing and distinguishing such effects in relaxor ferroelectrics on ARCS. For example, in our recent paper on PMN-PT relaxor ferroelectrics available here <http://advances.sciencemag.org/content/2/9/e1501814.full> (Manley, M. E. et al., *Science Adv.* **2**, e1501814 (2016)) a cut of the elastic scattering shows obvious signatures of the underlying inhomogeneity. The nanoregions produce strong diffuse patterns around the Bragg peaks and the local off centering of Pb atoms produces broad oscillations in the diffuse background (see Fig. 3 in this paper). These patterns were obtained using the same neutron scattering instrument (ARCS) on a similar size crystal. The complete absence of diffuse scattering of either type in our fresnoite crystal (Fig. 2) shows that it has at least three to four orders of magnitude lower intensity associated with static inhomogeneity than in the relaxors. Our crystal also does not contain any twins or low angle boundaries, as these would show up as an extra set of reflections in our three-dimensional single crystal diffraction patterns. Small angle micron size domains would result in diffraction peak splitting. Since no peak splitting is observed either the domains are not there or the tilt angles differences are too small to be resolved.

Comment #3 of Reviewer #1: *My key queries are: 1. Is there an independent verification of the stoichiometry? In particular, have chemical analyses been undertaken to ensure the Ba:Ti:Si = 2:1:2 stoichiometry is correct without Ba vacancies?*

Reply: Yes, a local chemical analysis company, Galbraith Laboratories, Inc., Knoxville, TN, verified the crystal stoichiometry using ICP mass spectrometry on a

small piece of material cut from the crystal used in the neutron scattering experiments.

Comment #4 of Reviewer #1: *2. Has TEM, high resolution backscattered SEM, or EBSD channeling of [001] sections been used to confirm the absence of micron sized domains.*

Reply: For our purposes single crystal neutron diffraction, which is sensitive to micron sized domains, is the most appropriate technique. The reason is that it is sensitive to the presence of domains anywhere within the entire crystal volume, the same volume in which the phonons and phasons are measured. Small angle micron size domains with small angular tilts will result in diffraction peak splitting since the different single-domain diffraction conditions will be tilted with respect to each other. Since no peak splitting was observed, either the domains are not there or the angle differences are too small to be resolved. A high density of stacking faults could be considered as creating domains, but as described above, diffuse elastic scattering was not observed. Micron sized twin domains are clearly ruled out by our three-dimensional single crystal diffraction patterns.

As a practical matter, measuring phonons in a crystal with a small mosaic from micron-sized domains does not change the result. From the point of view of a phonon/phason, a micron is a long distance. So a phonon or phason in a micron-sized domain is essentially identical to a phonon in a centimeter domain free crystal. Scattering from the domain walls only becomes important at low temperatures as the mean free paths of the phonons become large, but even in this case it would not show up in the measured phonons unless we used very high-energy resolution. Furthermore, if the Q resolution is not high enough to see the splitting in the Bragg peaks then it will not be resolved in the phonons or phasons either.

Understanding the level of defects does become important when we look at potential pinning of the phasons. It would be interesting in future work to add defects in a controlled way to modify phason pinning.

Comment #5 of Reviewer #1: *It would have been useful to see if the supersonic performance of the crystal was maintained after temperature cycling through the incommensurate transition temperature. Two cycles would be sufficient.*

Reply: Interesting question. We actually did cycle the temperature three times during the experiment and we observed no change in the phason dispersion. The fact that the phasons did not change means that either the defect structure did not change significantly or the phasons are not very sensitive to the small changes in the defect structure.

Comment #6 of Reviewer #1: *In general, mellites are more homogeneous than fresnoites and perhaps this could be commented upon.*

Reply: After reviewing the literature we suspect that the reviewer may have meant mullite rather than mellite (a.k.a. honeystone). Mullite has a very stable incommensurate superstructure that persists until melting (1840 °C), whereas we could not find any literature on incommensurate mellites. Mullite is also well matched to neutron scattering since it is composed of Al, Si, and O. Here are some references on the incommensurate structure of mullite:

[37] Desmond, J., McConnell, C. & Heine, V. Incommensurate structure and stability of mullite. *Phys. Rev. B* **31**, 6140(R) (1985).

[38] Angel, R. J. & Prewitt, C. T. The incommensurate structure of mullite by Patterson synthesis. *Acta Cryst.* **B43**, 116 (1987).

[39] Padlewski, S., Heine, V. & Price, G. D. A microscopic model for a very stable incommensurate modulated mineral: mullite. *J. of Phys: Cond. Matt.* **5**, 3417 (1993).

The obvious advantage of mullite as a candidate for studying phasons is that its incommensurate structure is stable all the way to melting. Hence, the contributions of phasons to thermal properties could be studied and utilized all the way to melting. The disadvantage is that its stability also suggests that the incommensurate modulation may not be as flexible as in fresnoite, which may limit what changes might be achieved with temperature and applied fields (stress, electric, etc.). We thank the reviewer for this suggestion.

In our revised manuscript we now include the following referenced sentence on mullite in the discussion section:

“At the other extreme, it would also be interesting to explore phasons in mullite, which has a very stable (rigid) incommensurate structure that persists to higher temperatures, all the way until the melting³⁷⁻³⁹.”

Reviewer #2

Comment #1 of Reviewer #2: *The manuscript by Manley et al. reports neutron scattering measurements of a fresnoite crystal, where branches of phasons were discovered as a result of the development of incommensurate superstructures below a transition temperature of 433K. By comparing the neutron scattering intensity maps above and below the transition temperature, the development of the incommensurate superstructure was clearly shown as emerging Bragg reflection points, around which dispersion cones of phasons were observed. Two interesting features were noted and discussed: the supersonic group velocity of phasons and the mismatch of the wavevectors between the phasons and the soft phonon mode. The former is of high interest to thermal energy transport, and the latter is explained by phason strain instability possibly induced by the piezoelectricity. The experimental observation of the phason dispersion is remarkable, and as far as this reviewer is concerned, the experimental evidence is solid and convincing. The finding of supersonic phasons is of*

interest to thermal transport, and is expected to stimulate follow-up investigations, for example how electric field can affect the phason properties. In this light I can recommend publication of this manuscript in Nature Communications, provided the following minor comments are addressed:

Reply: We thank the referee for this excellent summary of our work and address each of the five comments in detail below.

Comment #2 of Reviewer #2: *1. It would greatly benefit a general audience if more background information could be given in the introduction part, in addition to citing the results of Ref. 16, given that the concept of phason is unfamiliar to the thermal transport community. For example, what is the physical picture (lattice vibration modes) of a propagating phason? what determines the velocity of a phason, and why its velocity can exceed the sound velocity? How much energy can the phasons carry, in comparison to phonons?*

Reply: In our revised manuscript we include a discussion and calculations estimating how much heat is carried by the phasons (see new Fig. 5 and the description in the section “Analysis of the lattice phason properties”). We also include anisotropic thermal diffusivity data from the literature, revised Fig. 4c, which indicate a contribution by phasons that is roughly consistent with this estimate.

To understand the high velocities of the phasons we need to first explain how the underlying solitons can move faster than sound. A classic solution of supersonic solitons is that of the exponential or Toda lattice, which is explained here https://www.physics.uoguelph.ca/applets/Intro_physics/kisalev/java/toda/index.html.

A physical explanation for the supersonic speed of solitons is as follows: The speed of sound is based on linear response theory. At small amplitudes the phonons are the solutions when the forces are linearly proportional to the atomic displacements, but this is not the case with solitons. Solitons are nonlinear solutions that exhibit augmented force constants because of the large nonlinear compression of the lattice around the soliton. The force per unit displacement for atoms around the soliton can thus be higher than that experienced by the plane waves and the response times can be correspondingly faster. Another way to put it is that the large compression of the lattice around the soliton warps the local environment in a way that produces stronger interatomic forces that enable the solitons to travel faster than the conventional plane waves.

Hence phasons, which involve the sliding motions of solitons (Fig. 1d), can in theory have faster response times than the crystal lattice simply because of the augmentation of the forces by the soliton strains. There may be other contributing factors such as site atom mass differences and these will be explored in future work (e.g. by substituting Sr for Ba).

To clarify the importance of nonlinearity in augmenting the propagation velocities of solitons, the following two sentences were added to the first paragraph of the introduction:

“The speed of sound is based on linear response theory, but solitons are nonlinear modes that exhibit locally augmented forces because of a compression of the lattice around the soliton. Solitons can warp the local environment in such a way that enables them to travel faster than sound.”

Comment #3 of Reviewer #2: *“2. In addition to discussing the heat capacity in Fig. 4b, it'd be helpful to also look at the thermal conductivity data, if it is available, to see whether there are signs of contribution from the phasons.”*

Reply: Anisotropic thermal diffusivity data is now included as Fig. 4c in our revised manuscript. Comparing the direction with propagating phasons (a-axis) to the direction without phasons (c-axis) the difference is a 20% increase in the thermal diffusivity along the a-axis as the phasons form. This behavior in the thermal transport is similar to what was reported for charge-density-wave phasons at low temperatures and attributed to phasons of “rather large velocity” (see Ref. [31]).

Comment #4 of Reviewer #2: *“3. In Fig. 1c, the legend “regular soliton lattice” is not correctly displayed. It would be helpful to provide color bars for Figure 2 and Figure 3. In Figure 1f, the incommensurate modulation is not clearly seen; it would be educational to clearly mark the incommensurate rotations of polyhedral.”*

Reply: The legend in Fig. 1c has been fixed and color bars have been added to all image plots in Figs. 2 and 3 in the revised manuscript.

The incommensurate rotations of the polyhedral can only be approximated because they do not repeat exactly in the 3D structure (they can be described as periodic in higher dimensions – analogous to quasicrystals). Thomas Höche worked out many crystal approximants for the fresnoite structure in his PhD thesis. His PhD thesis is available online at http://www.iom-leipzig.de/fileadmin/redaktion/pdf/Dissertationen/Habilitation%20H%C3%B6che_sicher.pdf

For educational purposes we now indicate a particular subset of these rotations using yellow arrows in revised Fig. 1f. This subset was chosen from those described in Höche’s thesis because it shows in a clean way how the rings of five (odd number) linked polyhedral geometrically frustrates rotations, which leads to translations of some polyhedral and more complex incommensurate modulation patterns. Notice in the revised figure reproduced below how propagating a rotation around a ring of five polyhedral near the bottom center forces a translation of one of the Si pyramids.

Comment #5 of Reviewer #2: 4. Line 142-145, it is unclear why the slowdown of the growth of the *t*-phason intensity with decreasing temperature below 400K indicates the *t*-phason is partly gapped. More explanation and background information would be helpful.

Reply: Below the transition the elastic peak intensity grows, but it is not directly correlated with the growth of the phason intensity (Fig. 3, g & h). If the phasons were perfectly gapless (ideal), then we would expect that the intensity of the phason would follow the intensity of the incommensurate reflection. This is because the phason structure factor correlates with the incommensurate reflection. The disconnection between the phasons and incommensurate reflection intensities implies that the connection is more complicated. This is not surprising since a phason that is truly gapless everywhere requires a perfectly incommensurate modulation. Any imperfections in the crystal can act as pinning centers for the phasons, thereby producing a small gap. However, since the crystal is not perfect the pinning may be inhomogeneous at the microscale. Meaning that the amount of gapping and at what temperature the gap may close to produce an elastic peak likely varies from place to place within our macroscopic crystal, and this gets averaged together in our measurements. We recognize the phrase “partly gapped” is confusing, so this word has been removed in the revised manuscript.

Comment #6 of Reviewer #2: 5. It is noted in the introduction that previous measurements of phasons suffered from damping that broadened the mode. It would be helpful to also discuss the role of phason damping in this work.

Reply: We do not observe phason damping directly in the neutron scattering measurements because the observed phason line width is resolution limited. We do,

however, expect that phason damping becomes important at long wavelengths near the incommensurate reflection.

An upper limit on the energy where the overdamping becomes important can be estimated from the temperature of the upturn in the heat capacity. This is because the upturn is either from damping or from a small gap. If it is caused by a gap and the damping is not yet observed, then the damping effect must occur at an even lower temperature. The upturn in the C_p/T^3 occurs below about 3 K, which corresponds to energy of about 0.27 meV. Using the mode density of states evaluated from the T-phason dispersion curve it is possible to calculate the ratio of the number of states between 0 and 0.27 meV and between 0 and 6 meV. This ratio is 2.7×10^{-8} based on our fit to the dispersion data shown in Fig. 5. Hence, the damped phasons at long wavelengths are not a factor in the thermal conductivity. This is now mentioned in the revised manuscript.

Reviewer #3

Comment #1 of Reviewer #3: *This is an interesting paper that reports on the observation of a phason mode in the piezoelectric mineral fresnoite by inelastic neutron scattering (INS) experiments. The standard theory of displacive incommensurate (IC) phase transitions as it has been formulated nearly 40 years ago, goes as follows: at some point a phonon mode becomes critical and softens at an IC reciprocal space position to finally condense at a given temperature. Below the critical temperature a new periodicity appears and the soft mode is further decomposed into modes that account for variations of the phase and of the amplitude of the modulation. Nevertheless the occurrence of this decomposition soft phonon \rightarrow phason + amplitudon (as nicely shown in the cartoon of Figs 1d and 1e) is not often observed in INS experiments for reasons that are still under debate. It is always nice to have one more compound where these ideas can be further tested, and I congratulate the authors for this new study.*

Reply: We thank the referee for this excellent summary of and for the positive comments on our neutron scattering measurements.

Comment #2 of Reviewer #3: *This compound offers at least two interesting features readily highlighted by the authors : (1) the transition temperature of this IC modulation is rather high, 433K, and (2) the group velocity of the phason mode is larger than the normal longitudinal speed of sound by a factor of 3. Implicit in this statement is the fact that because of the presence of this IC structure, and thus of phasons with "supersonic" velocities, in fresnoite at high temperature this property can be used to dissipate thermal energy more efficiently and overcome the limits of normal acoustic phonons. It is here where this reviewer thinks that the authors have made some conceptual mistakes and where the experimental support of their hypothesis is terribly lacking.*

Reply: This is a really good point that more supporting data and explanation are needed in the manuscript. In our revised manuscript we now include analysis of anisotropic thermal diffusivity measurements in revised Fig. 4c and calculations of factors contributing to thermal conductivity in new Fig. 5. We also include new Ref. [31], which shows the contribution of charge-density-wave phasons to thermal conductivity at low temperatures. To emphasize the role that phason might play in thermal logic devices (see Ref. [1]), we also discuss a simple way that ballistic supersonic phasons might be used to make a thermal analog of a surge protector in a small-scale device. In this case, it is the supersonic nature of the phasons, and not their heat capacity that is most central.

The measured data (Fig. 4) shows that the phasons increase the thermal diffusivity (and therefore thermal conductivity) in the in-plane direction by about 20% at room temperature, and this will clearly grow on cooling. Because the phasons do not contribute to the out-of-plane *c*-axis direction there is also an associated increase in the anisotropy in the thermal diffusivity from 1.8 at 870 K to 2.7 at room temperature. Hence, the phasons help to spread heat within the basal plane of the crystal more effectively.

To explain the phason contribution to diffusive transport at room temperature, in our revised manuscript we analyze the Boltzmann transport equation components of the thermal conductivity tensor:

$$\kappa_{\alpha\alpha} = \frac{1}{V} \sum_{q_s} C_{q_s} v_{\alpha q_s}^2 \tau_{\alpha q_s}, \quad (1)$$

where $v_{\alpha q_s}$ is the group velocity of the mode q_s , $\tau_{\alpha q_s}$ is the mode lifetime for transport in direction α and C_{q_s} is the mode specific heat. At high temperatures ($k_B T \gg \hbar\omega$) the heat capacity of each acoustic phonon contributes k_B per primitive cell. There are two transverse and one longitudinal acoustic phonons making for a total of $3k_B$ per primitive cell. There are an additional $3k_B$ for each additional atom in the primitive basis, and this heat is carried by optical phonons. Therefore, on a per atom basis, the number of atoms in the primitive cell divides the contribution of the acoustic modes by the number of atoms in the primitive cell. The same accounting rules apply to the modes of the incommensurate superstructure, except the larger number of atoms in the super cell divides each degree of freedom. In the basal plane the superstructure is incommensurate with reflections near $h \sim 0.4$, which corresponds to a cell length increase of about 2.5. Therefore the cell volume increases in the plane by $\sim 1 \times 2.5 \times 2.5 = 6.25$, which means there are 6.25 times more atoms in the supercell than in the original primitive cell. It follows that the phason contribution to the specific should be about $1/6.25$ (16%) of the contribution from the acoustic phonons. However, the thermal conductivity terms in eqn. (1) also scale as the square of the group velocities, so the high phason group velocity contributes a large factor in favor of the phasons. New Fig. 5 (reproduced below) shows that, based on the fits to the dispersion curves (Fig. 5a), the smaller T-phason density of

states (Fig. 5b) when multiplied by the higher phason group velocities (Fig. 5c) results in a contribution that is similar to the TA phonon (Fig. 5d). The density-of-states weighted square velocity integrates to an area (blue region in Fig. 5d) that is about 50% that of the TA phonon (grey region in 5d). This means that for similar excitation lifetimes the high-temperature T-phason contribution to thermal conductivity is expected to be $\sim 50\%$ that of the TA phonon. It is therefore not surprising that the phasons contribute $\sim 20\%$ of the in-plane thermal conductivity at room temperature (Fig. 4c). This analysis neglects unknown differences in the mode lifetimes. The acoustic phonons and phasons have measured line widths that are instrument resolution limited, so we cannot resolve any difference in lifetime broadening. The known effect of phason damping at long wavelengths is not significant, however, because the contribution of long wavelength modes to the density of states is inherently small in three dimensions.

Figure 5 from revised paper.

A more formal version of this description has been added to the “**Analysis of the lattice phason properties**” section in our revised manuscript.

While the phason contribution to bulk transport is important, a more direct application would be to use supersonic phasons as a signal in the design of smart heat transport in small-scale devices. Consider, for example, a device with two components separated by a supersonic phason carrying insulator of a thickness less than the mean free path of the phasons/phonons. If one component suddenly heats up, energy in the form of ballistic phonons and phasons will advance towards the other component. Because the fastest phasons propagate at speeds many times faster than the speed of sound they will arrive at the other component first, providing an advanced warning signal that a larger heat pulse is about to arrive. This thermal pre-signal could then be used to trigger a protective measure ahead of the advancing phonon heat front, perhaps by pre-triggering an insulator to metal transition in a layer of vanadium dioxide as an electrical switch. This thermal analog of a surge protector would exploit directly the supersonic propagation of phasons.

Research on such thermal logic devices is a rapidly emerging field of research, as discussed in Ref. [1], and supersonic phasons could play an important role in speeding up ballistic processes.

This argument about supersonic phasons used as an ingredient of a thermal logic device has been added to the “**Discussion**” section in our revised manuscript.

Comment #3 of Reviewer #3: *1.- The manuscript reports inelastic neutron scattering and not thermal conductivity experiments. Without these latter experiments the conclusions announced from the title of this manuscript can not be supported and remains very speculative. In addition, as I detail below, there are good reasons to believe that phasons do not serve for lattice energy transport purposes.*

Reply: For reasons discussed above we disagree with the argument that phasons do not serve for lattice energy transport purposes. Our revised manuscript now includes an analysis of thermal transport measurements from the literature (see revised Fig. 4) and an analysis of the terms in the transport equation (see Fig. 5), both indicating that the phasons make an important contribution to bulk transport. Reference [31] also demonstrated previously that CDW phasons of “rather large velocity” contribute to thermal conductivity at low temperatures. We also speculate that in small-scale devices ballistic supersonic phason transport could be used to make novel thermal logic components for smarter thermal management. This qualitatively different way of thinking about thermal management is a rapidly emerging field of research (see Ref. [1]).

Comment #4 of Reviewer #3: *2.- The occurrence of a phason mode in this compound is very likely beyond dispute, at least this reviewer does not dispute it. However it has not been experimentally shown whether the phason mode is gapped or gapless. In page 5 the authors claim it is gapped (I do not understand the term “partly gapped”) and a few lines below they hold that the gap closes at lower temperatures (!). No experimental evidence is provided for any of these assertions. The authors know that if they want to properly assess the phason gap condition and thus help disentangle this problem INS experiments with cold neutrons, that warrant a better energy resolution, are mandatory. Experiments in different IC compounds have shown that the phason mode always (?) has a gap or at least it is overdamped (like in K₂SeO₄). I believe the authors are acquainted with the book by Binc and Leyanyuk “IC phases in dielectrics” where they can find an early account of the phason gap features in several compounds.*

Reply: We agree that below the transition temperature where incommensurate reflections appear we cannot observe a gap directly. However, above this temperature the phasons appear without a corresponding incommensurate reflection (Fig. 3). A truly gapless phason would result in a reflection at the point where the phason touches the elastic line since the amplitude of a mode diverges at zero frequency. In this sense, the phasons at least begin gapped above the transition

temperature. This is not something that would normally be observed because the phason usually forms below the soft phonon mode.

As described earlier, below the transition the elastic peak intensity grows, but it is not directly correlated with the growth of the phason intensity (Fig. 3, g & h). If the phasons were perfectly gapless, then we would expect that the intensity of the phason would follow the intensity of the incommensurate reflection. This is because the phason structure factor correlates with the incommensurate reflection. The disconnection between the phasons and incommensurate reflection intensities implies that the connection is more complicated. This is not surprising since a phason that is truly gapless everywhere requires a perfectly incommensurate modulation. Any imperfections in the crystal can act as pinning centers for the phasons, thereby producing small gaps. Since the crystal is not perfect the pinning may be inhomogeneous at the microscale. In this case the gap may be “partial” in the sense that it likely varies from place to place within our macroscopic crystal, and this gets averaged together. We recognize the phrase “partly gapped” may be confusing to some readers. This word has been removed in the revised manuscript.

In the heat capacity, it is not possible to say whether the upturn is all a damping effect or if there might be a small gap at low temperatures. A gap much smaller than 3 K (0.27 meV) would not be detected without lower temperature heat capacity measurements. To clarify this point we added “or possibly a small gap from phason pinning” in the description of heat capacity.

Comment #5 of Reviewer #3: *As the authors have stated the C_p/T^3 plot shows an upturn at low temperatures, although there is no clear signature of a phason gap, in much the same way as the results displayed in Fig. 4. This upturn is very likely produced by the damping of the phason gap. A recent account of this kind analysis in IC ThBr₄ can be found in Remenyi et al., PRL 114, 195502 (2015), following the ideas in ref. 30. The presence of a phason gap and damping hinders the thermal conductivity. In view of this and of the presence of an upturn of the CP/T^3 attributed to phason damping, this reviewer does not understand why the authors still believe the title of this manuscript!*

Reply: As can be seen in our new Fig. 5, the phasons at low energies/long wavelengths contribute practically nothing to the high temperature bulk thermal conductivity. The physical reason is that the density of states of long wavelength modes is negligibly small compared to the shorter wavelength modes.

The upturn in the C_p/T^3 occurs below about 3 K, which corresponds to energy of about 0.27 meV. Using the mode density of states evaluated from the T-phason dispersion curve, it is possible to calculate the ratio of the number of states between 0 and 0.27 meV and between 0 and 6 meV. This ratio is 2.7×10^{-8} based on our fit to the dispersion data shown in Fig. 5. Hence, the damped phasons at long wavelengths are not a factor in the thermal conductivity. This is now mentioned in the revised manuscript.

A gap at very low energies, also does not contribute much to the high temperature bulk thermal conductivity for the same reason.

Hence, the expected phason damping at long wavelengths is not an important factor in the high temperature thermal conductivity. It may become more important at low temperatures as the higher energy states become depopulated. However, judging from the upturn in C_P/T^3 this only becomes important at very low temperatures.

Comment #6 of Reviewer #3: *3.- In order for the phason to be relevant for lattice transport properties an order of magnitude of its contribution to the density of states should be provided. The crucial point is to understand that the phason and amplitudon modes just appear in the close neighbourhood of the IC position, and that the volume thus encapsulated is relatively small compared to the volume of the Brillouin zone. Put in a different way, if there are $3N$ vibration modes, N the number of atoms in the unit cell, each mode contributes as 1. In the case of phasons this number should be much smaller, in many cases its contribution falls down to 0.01-0.1. You can check this issue in, for instance, the paper by Boriack & Overhauser, PRB18, 6454 (1978). This argument on the smallness of the volume of the phason excitation stands irrespectively of whether the phason is gapped or gapless.*

Reply: This is a good point. We have addressed it in the new Fig. 5 in our revised manuscript. Please see the reply to Comment #2 of Reviewer #3 above for details. Thank you.

Comment #7 of Reviewer #3: *3a.- If the phason is gapped one should expect a deviation of the C_P/T^3 curve, hardly visible because of the smallness of its contribution. There has been some heated discussions in relation to the contribution of the phason to the low temperature properties of the heat capacity. I wonder if the authors are acquainted with this. Figure 4 shows specific heat data with a phonon calculation unable to fit the data below 10 K. May be this a gapped phason contribution?*

Reply: While there may be a small gap at low temperatures, it can neither be confirmed nor ruled out from the heat capacity data in Figure 4. In our revised manuscript we now mention both damping and possibly an unresolved gap contributing to the upturn in C_P/T^3 . If it were just a gap, then at some sufficiently low temperature the curve would be expected to turn back down. If this does occur it would have to be below 2 K (0.18 meV), indicating that any gap implied from heat capacity would have to be very small compared to the phason energy cut off at 6 meV.

As discussed by Bruce and Cowley in Ref. [28] there are many ways that phasons may become pinned and therefore gapped, including solitons preferentially sticking to specific sites in the unit cell or structural defects. So it would not be surprising if there were a small gap even at low temperatures. As mentioned earlier, the gap

could even vary from place to place within the crystal. However, this would not change the main conclusions of our paper.

Comment #8 of Reviewer #3: *3b.- If the phason is gapless one should expect that its contribution to C_p goes to slightly decrease (!) the Debye temperature from regular acoustic phonons. And again this correction should be very very small. Again it should be interesting to put some numbers on the renormalisation of the Debye temperature by gapless phasons.*

Reply: At low temperatures each mode contributing to C_p/T^3 adds a constant term that scales like f_j/θ_j^3 , where θ_j is the effective Debye temperature of the j th gapless mode and f_j accounts for the fractional weighting of the mode in the density of states. Since $\theta_j \propto v_j$ (mode group velocity) and the group velocities of the phasons are 3 to 4 times larger, these factors when cubed in the inverse will be very small indeed. Note that $1/3^3 = 0.037$ and $1/4^3 = 0.015$. The weighting factor for the fraction of the density of states is also small (~ 0.16), resulting in terms that are less than 1% the size of the terms from the acoustic phonons in the heat capacity. Hence, these terms can be neglected and the only factor that really matters when calculating the decrease in the effective Debye temperature is the decrease in the acoustic mode contribution to the density of states owing to the reallocation of states to the phasons. By our estimate this is about 16%. However, because of long wavelength phason damping this contribution does not really become a constant and instead shows an upturn (Fig. 4b).

At high temperatures, the specific heat tends to a constant as all states become filled. This value is unaffected by whether there are phasons or not since it just accounts for all the degrees of freedom in the system, which for lattice excitations amounts to all the atoms contributing $3k_B T$ each.

At intermediate temperatures the phason contribution is somewhere between the LA and TA phonons and we did not find that this changes the resulting fit to the specific heat in any appreciable way. This is because at intermediate energies, 3 meV to 6 meV, the phasons appear between the TA and LA phonons and are also weaker.

Comment #9 of Reviewer #3: *4.- Most of the heat transport is conducted by transverse acoustic phonons at all temperatures. The reason for that is that the heat transport is proportional to the phonon density of states (PDOS) and the transverse acoustic phonons have a larger PDOS than the longitudinal ones. Any "new" mode with larger group velocity should renormalise the PDOS, and thus the Debye temperature, to lower values. I am afraid that this has not been checked either.*

Reply: While we agree that the phonon DOS should be renormalized by the phasons, extracting the effect on the acoustic phonons from a powder measurement of the phonon DOS is not so trivial. We know because we tried. The problem is that as the phasons form the soft optic mode rises up in energy thereby removing intensity

from the same energy range where phasons add intensity. These competing contributions to the low energy phonon DOS can be seen by inspecting the phonons and phasons in Figure 2. At 485 K there is a soft phonon mode that dips down to ~2 meV while the phasons are weak. At 350 K the phasons appear stronger while the soft phonons appear to retreat up to above 3 meV. Hence, when viewed from the phonon DOS, which integrates over reciprocal space, the two effects tend to cancel each other and make separation practically impossible.

Comment #10 of Reviewer #3: *5.- Thermal conductivity (TC) is rather difficult to calculate from first principles. On general grounds the TC evolution as a function of temperature goes as follows. At low temperatures, when the boundary scattering mechanism dominates, the conductivity follows the expected increase as T^3 . As the temperature increases, mass-defect scattering becomes important and reduces the T^3 dependence. Three-phonon scattering sets in at finite temperatures, becoming dominant at high temperatures. As a result, the conductivity reaches a maximum (the position of the maximum goes from 20K in Si to 100K in C-diamond) before starting to decrease. There are four criteria for choosing high-thermal-conductivity materials: (i) low atomic mass, (ii) strong interatomic bonding, (iii) simple crystal structure, and (iv) low anharmonicity. Conditions (i) and (ii) help increase the Debye temperature, condition (iii) means a low number of atoms per unit cell, resulting in fewer optical branches and hence fewer anharmonic interactions, and condition (iv) means reduction in the strength of anharmonic interactions. This compound may satisfy the first, hardly the second criteria but not at all the third and fourth.*

Reply: Our objective here is not to optimize the thermal conductivity by applying well-known criteria, but to understand an additional criterion to improve thermal conductivity and possibly enable new kinds of thermal management tools, such as a thermal analog of a surge protector (see response to Comment #2 above). The fact that fresnoite is also a good piezoelectric sensor enriches the possibilities. It would be interesting for future research to merge these criteria with a phason mechanism to optimize thermal conductivity.

Comment #11 of Reviewer #3: *Summarising, the manuscript contains important information on IC dynamics, in particular the wavevector rotation away from the soft mode position, that is worth publishing. However this reviewer can not recommend this manuscript for publication in Nature Communications. Indeed, each one of these above mentioned 5 points invalidates the authors view on the possibility of supersonic transport of the thermal energy of lattice vibrations by phasons, which is the message that would warrant publication in this journal.*

Reply: We hope we have convinced the reviewer that lattice phasons do indeed enhance the thermal conductivity of fresnoite at ambient conditions. Revised Fig. 4c shows the contribution in data from the literature and new Fig. 5 explains why it is there. We also hope that readers will appreciate that supersonic lattice phasons may find applications in the emerging science of thermal logic devices as reviewed in Li et al. Colloquium: Phononics: Manipulating heat flow with electronic analogs and

beyond. *Rev. Mod. Phys.* **84**, 1045 (2012). As discussed in this review, lattice energy need not only be treated as waste but can be used to carry and process information. In this context the relatively high propagation velocities of the phasons seem particularly promising. We suggest, for example, that ballistic supersonic phasons traveling over short distances might be utilized as an advanced thermal signal in a thermal analog of a surge protector.

Reviewers' comments:

Reviewer #1 (Remarks to the Author):

The authors have answered my queries, and those of the other reviewers, satisfactorily.

Unfortunately I caused confusion by typing 'mellite' rather than 'melilite' which another incommensurate structural family closely related to fresnoite. I would be pleased if the authors were to add a short comparative assessment of the likelihood of these phases showing phason-mediated lattice transport.

I believe the outcomes of this research are of sufficient novelty to warrant publication in Nature Communications.

Reviewer #2 (Remarks to the Author):

The authors have addressed my comments satisfactorily, and I can recommend publication of this manuscript as it is.

Reviewer #3 (Remarks to the Author):

The authors have performed outstanding work in answering most of the referees' queries and in improving their manuscript beyond expectations. I have to congratulate them for this. Their new analysis and quantification of the heat transport in the presence of phasons is right on target. As a result the phason contribution is estimated to enhance the heat transport by 20%. However this analysis pertains to low-medium temperature transport not to the high temperature (above 100K) where the heat transport is mostly determined by multiphonon scattering processes. The quantification of the multiple scattering has not been attempted and therefore it is not possible to quantify the utility of this compound for high temperature transport, regardless of the effect of the phason branch. 20% of something that I presume to be small is going to be small. I have pointed this out in comment #10 to which the authors' response is "Our objective here is not to optimise the thermal conductivity...". We can debate about that statement but in the absence of actual thermal conductivity measurements the answer to the question of whether or not phason dynamics can be used to enhance thermal conductivity at low and high temperatures is yes it can, but the amount is very likely to be irrelevant for applications. You are always better off with well known and characterised compounds such as diamond, graphite, SiC, BAs, Si, etc. To this reviewer answering this question is at the core of this manuscript as it appears in the appealing, though misleading, title: Supersonic transport of lattice energy by phasons.

Another point that the authors make in their manuscript is the challenge in energy sciences to control the heat carried by lattice vibrations (first statement of the manuscript page 2). This point is further retaken in the Discussion (first paragraph, a whole paragraph !) by explaining how ballistic supersonic phasons would be applied to use them as a signal in a thermal logic device. This is nice, it may justify a project to find money to complete some prototypes but it is well beyond the scope of a neutron scattering work.

To summarise, I cannot do anything more with this manuscript. It remains a nice and solid piece of work, well written and structured albeit a pointless exercise in the absence of proper data.

Re: NCOMMS-17-23667B
Response to reviewers:

Reviewer #1

Comment of Reviewer #1: *“The authors have answered my queries, and those of the other reviewers, satisfactorily.*

Unfortunately I caused confusion by typing 'mellite' rather than 'melilite' which another incommensurate structural family closely related to fresnoite. I would be pleased if the authors were to add a short comparative assessment of the likelihood of these phases showing phason-mediated lattice transport.

I believe the outcomes of this research are of sufficient novelty to warrant publication in Nature Communications.”

Reply: We thank the reviewer for constructive criticisms and helpful comments, which were valuable in revising our manuscript.

Melilites are indeed interesting. Natural melilite has an incommensurate modulation that develops below 359 K – similar to the 433 K for fresnoite – but differs in that the incommensurate modulation wave vector is temperature dependent¹. This is interesting from the point of observing the phasons since the key to observing clear phasons in fresnoite is that they separate from the soft phonon mode. Melilites, like fresnoites, are also piezoelectric², which suggest that both temperature and electric field could be used to control the phasons.

1. Bindi L., & Bonazzi, P. Incommensurate-normal phase transition in natural melilite: and in situ high-temperature single-crystal study. *Phys. Chem. Minerals* **32**, 89 (2005).
2. Shen, C., Zhang, S., Cao, W., Cong, H., Yu, H., Wang, J. & Zhang, H. Thermal and electromechanical properties of melilite-type piezoelectric single crystals. *J. Appl. Phys.* **117**, 064106 (2015).

The structural similarity of the melilites to the fresnoites suggests that the phason contribution to thermal transport is probably similar. However, the contribution depends sensitively on details of the phason dispersion curves.

In our revised manuscript we now add the references above and the following sentence in the discussion mentioning melilite:

“Melilite, a related piezoelectric mineral with a similar incommensurate modulation³⁷, has modulation wave vectors (phasons) that vary linearly with temperature³⁸.”

This is mentioned just above the previously added comment on mullite. The two cases are interesting to compare with fresnoite. While the incommensurate modulation in mullite is the most and persists all the way to melting, the modulation in melilite is more flexible in the way it moves with temperature, but is somewhat less stable at high temperatures (395 K for melilite versus 433 K for fresnoite).

Reviewer #2

Comment of Reviewer #2: *"The authors have addressed my comments satisfactorily, and I can recommend publication of this manuscript as it is."*

Reply: We thank the reviewer for constructive criticisms and helpful comments, which were valuable in revising our manuscript.

Reviewer #3

Comment #1 of Reviewer #3: *"The authors have performed outstanding work in answering most of the referees' queries and in improving their manuscript beyond expectations. I have to congratulate them for this. Their new analysis and quantification of the heat transport in the presence of phasons is right on target. As a result the phason contribution is estimated to enhance the heat transport by 20%."*

Reply: We thank the reviewer for the kind words and for the constructive criticisms and valuable comments, which were of great help in revising the manuscript.

Comment #2 of Reviewer #3: *"However this analysis pertains to low-medium temperature transport not to the high temperature (above 100K) where the heat transport is mostly determined by multiphonon scattering processes. The quantification of the multiple scattering has not been attempted and therefore it is not possible to quantify the utility of this compound for high temperature transport, regardless of the effect of the phason branch. 20% of something that I presume to be small is going to be small. I have pointed this out in comment #10 to which the authors' response is "Our objective here is not to optimise the thermal conductivity...". We can debate about that statement but in the absence of actual thermal conductivity measurements the answer to the question of whether or not phason dynamics can be used to enhance thermal conductivity at low and high temperatures is yes it can, but the amount is very likely to be irrelevant for applications. You are always better off with well known and characterised compounds such as diamond, graphite, SiC, BAs, Si, etc. To this reviewer answering this question is at the core of this manuscript as it appears in the appealing, though misleading, title: Supersonic transport of lattice energy by phasons."*

Reply: Fresnoite does not exhibit phasons at temperatures >1000 K. However, in the discussion we did note that, ***"it would also be interesting to explore phasons in***

mullite, which has a very stable (rigid) incommensurate structure that persists to higher temperatures, all the way until the melting³⁷⁻³⁹.

With the word “transport” in the title we meant it in the literal sense of moving something from one place to another. Obviously, this is not the whole story for thermal conductivity, which is sometime referred to interchangeably with thermal transport. To avoid confusion we have changed the title to:

“Supersonic propagation of lattice energy by phasons in fresnoite”

The use of propagation here means the transmission of lattice energy along a particular direction. In our neutron scattering measurements we measure the propagation velocity of phasons along particular crystallographic directions, so this is a more appropriate title.

Comment #3 of Reviewer #3: *“Another point that the authors make in their manuscript is the challenge in energy sciences to control the heat carried by lattice vibrations (first statement of the manuscript page 2). This point is further retaken in the Discussion (first paragraph, a whole paragraph !) by explaining how ballistic supersonic phasons would be applied to use them as a signal in a thermal logic device. This is nice, it may justify a project to find money to complete some prototypes but it is well beyond the scope of a neutron scattering work.”*

Reply: The first paragraph in the Discussion has been removed. In its place we provide the following single sentence at the end of the thermal conductivity analysis section:

“While the phason contribution to bulk transport is important, a potentially more interesting application would be to use fast phasons as a signal in a thermal logic device¹, although it remains to be seen whether phason detection can be made practical.”